Subject Area:
developmental biology/genetics

Keywords:
*Drosophila*, muscle, microCT, ageing, DLM

Author for correspondence:
Dhananjay Chaturvedi
e-mail: dhananjayc@ncbs.res.in

# Adult *Drosophila* muscle morphometry through microCT reveals dynamics during ageing

Dhananjay Chaturvedi[1], Sunil Prabhakar[2], Aman Aggarwal[1,3], Krishan B. Atreya[1] and K. VijayRaghavan[1]

[1]National Center for Biological Sciences, and [2]microCT and EM Facility, National Center for Biological Sciences, TIFR, GKVK Campus, Bellary Road, Bengaluru 560065, India
[3]Manipal Academy of Higher Education, Manipal, Karnataka 576104, India

DC, 0000-0002-3957-1236; SP, 0000-0002-3810-7976; AA, 0000-0001-8606-3368; KBA, 0000-0002-9676-5862; KV, 0000-0002-4705-5629

Indirect flight muscles (IFMs) in adult *Drosophila* provide the key power stroke for wing beating. They also serve as a valuable model for studying muscle development. An age-dependent decline in *Drosophila* free flight has been documented, but its relation to gross muscle structure has not yet been explored satisfactorily. Such analyses are impeded by conventional histological preparations and imaging techniques that limit exact morphometry of flight muscles. In this study, we employ microCT scanning on a tissue preparation that retains muscle morphology under homeostatic conditions. Focusing on a subset of IFMs called the dorsal longitudinal muscles (DLMs), we find that DLM volumes increase with age, partially due to the increased separation between myofibrillar fascicles, in a sex-dependent manner. We have uncovered and quantified asymmetry in the size of these muscles on either side of the longitudinal midline. Measurements of this resolution and scale make substantive studies that test the connection between form and function possible. We also demonstrate the application of this method to other insect species making it a valuable tool for histological analysis of insect biodiversity.

## 1. Introduction

Fruit flies achieve remarkable wing beat frequencies (approx. 200 Hz). The beating motion of wings is created by a composite muscle apparatus in the *Drosophila* thorax. The power stroke arises through the function of two sets of indirect flight muscles (IFMs). These sets of muscles, the dorsal longitudinal muscles (DLMs) and dorso-ventral muscles (DVMs), are positioned at an angle with respect to each other and connected to the exoskeleton through tendons. Their asynchronous contraction cyclically deforms the thorax along the longitudinal and dorsoventral axes. This cyclical thoracic contraction is translated into the beating of wings [1].

IFMs are the only reported large *Drosophila* muscle group that shares myofibrillar architecture with mammalian skeletal muscles [2]. Despite significant differences in modes and frequency of activation from mammalian counterparts, they have been exceedingly informative in understanding *in vivo* principles of muscle development such as myotendinous junction development, nerve muscle interaction and sarcomere formation [3–7]. Both DVMs and DLMs attain stable structures post-pupariation that are maintained throughout the life of fruit flies. Therefore, they are ideal to model homeostatic muscle tissue *in vivo*. Owing to the relatively short lifespan of fruit flies, IFMs have also been used to model the effects of ageing on muscle function [8,9]. A consistent decrement in wing beating frequency and flight duration has been observed [10]. We aimed to analyse what morphological changes in muscles may be contributing to this observation. For this specific study, we have limited our analysis to the subset of IFMs called the DLMs.

royalsocietypublishing.org/journal/rsob   Open Biol. **9**: 190087

The depth of visualizing adult *Drosophila* muscle morphology is limited by technical challenges such as antibody penetration and tissue clarity. Often, circumventing these issues through clearing agents leads to tissue shrinkage. Therefore, confocal microscopy protocols find limited applications for *in situ* imaging of whole adult muscle tissues.

Muscle anatomy studies rely on staining dissected preparations, which inevitably changes the morphology of tissue to some degree. Specifically, in genetic conditions where DLMs are fragile, accurate morphometry *in situ* is impossible. Yet, these conditions are probably better indicators for comparisons such as adult-onset human muscle pathology. Accurate, quantitative and non-invasive measurement of adult *Drosophila* musculature is therefore much needed.

Micro-level computed tomography (microCT) analysis bridges resolution and depth of visualization. Tissue repair and degeneration studies demand three-dimensional visualization of tissue *in situ*. To this end, various fixation methods and contrasting agents have been applied in various tissues. Crucially, several sample preparation methods involve ethanol and high salt concentrations for fixation and storage of samples [11–13]. The disparity in DLM morphologies between these preps and immunohistochemical preps is obvious. To address this discrepancy, we successfully adapted staining with Lugol's solution in phosphate buffer saline, to retain morphology as observed with conventional immunohistochemistry protocols for DLMs.

In *Drosophila* cohorts of both sexes, we have found consistent changes in volume and fascicle arrangement within individual DLMs with age. Total DLM volumes in males and females increase at least up to 28 days post-eclosion. DLMs on opposite sides of a thorax are likely to have dissimilar volumes. Also, specific DLM fibres grow in volume during this period differently in females compared with males.

# 2. Material and methods

## 2.1. *Drosophila* husbandry

Wild-type *Canton S* flies were grown on cornmeal agar at 25°C on a 12 L : 12 D cycle. For accurate and representative samples of ageing, cohorts of animals were collected within an hour post-eclosion.

Twenty animals per vial were grown in the abovementioned conditions. Males and females were grown in separate vials. The number of females in our dataset at days 2, 7, 14 and 28 were 14, 17, 18 and 14 animals, respectively. The number of males in our dataset at days 2, 7, 14 and 28 were 15, 16, 21 and 16 animals, respectively.

$sply^{05091}$/Df(2R)BSC433 animals were generated by crossing $sply^{05091}$/CyO-GFP (derived from BDSC stock no. 11393) and Df(2R)BSC433/CyO-GFP parents (derived from BDSC stock no. 24937). First instar larvae were selected by the absence of the CyO-GFP balancer, transferred to cornmeal agar vials, and grown without their heterozygous siblings. Df(2R)BSC433/+ animals were generated by crossing Df(2R)BSC433/CyO-GFP with $w^{1118}$. Five-day-old females were used for this study ($n = 3$ per group).

## 2.2. Sample preparation for microCT scanning

At appropriate time points, groups of flies were anaesthetized on ice and transferred to 4% paraformaldehyde (PFA) made in PBS in a dish. Thoraces were dissected out by pulling out heads and cutting away abdomens and legs while retaining the wings. Thoraces were then transferred into a tube with 300 µl of the fixative solution, making sure all samples were submerged. These samples were incubated at room temperature for 3 h with gentle shaking. Subsequently, fixative was aspirated out and discarded followed by two 15 min washes at room temperature with 1 ml PBS per tube. 200 µl of staining solution—1% elemental iodine (1.93900.0121, Emparta, Merck) with 2% potassium iodide (no. 15 724, Qualigens) dissolved in PBS—was then added to each tube, making sure all thoraces were submerged.

Samples were incubated in staining solution with gentle shaking at room temperature overnight.

## 2.3. microCT scanning

Prior to scanning, each thorax was washed twice in 500 µl of PBS for 15 min at room temperature. Each thorax was dipped in paraffin oil. This step is critical to retain moisture and iodine in tissue during the scan. Individual thoraces were then mounted on a micro-positioning stage tipped with petroleum jelly, and wings were used to position and stabilize the thorax. These samples were then scanned on Bruker Skyscan-1272 at 40 kV, 250 µA, 4940X3280 pixels, averaging set to 2, without filters at 0.5 µm resolution with a rotation step of 0.55° for 180°. Each scan takes a total of 40 min.

## 2.4. Data processing and volume calculation

From each sample, all projection images were imported to NRecon software (Bruker Instruments) for a three-dimensional reconstruction with 5 unit smoothing.

The three-dimensional resolution of each CT virtual section is undersampled by a factor of 4, to speed up computation. The stack was reoriented in DATAVIEWER (Bruker) to align with coronal and sagittal planes. Regions of interest (ROI)-specifying muscles were drawn on this reoriented stack in CTAN (Bruker). Thresholding on the stack was done using the automatic two-dimensional Ridler–Calvard method, and volumes of signal bearing voxels within the marked ROI were calculated from these binarized images in CTAN. Statistical significance was calculated through the two-tailed Wilcoxon rank sum test in MATLAB.

Graphs in figure 3 were plotted using PYTHON. Volume maps in figure 2 were manually drawn on Adobe ILLUSTRATOR. The side length of each square was calculated in the following way. Briefly, to represent a volume $V$ as a square of side $s$, square roots of individual muscle volumes were calculated,

$$s^2 = bV \;\Rightarrow\; s = \sqrt{(bV)},$$

where $b$ is a constant of appropriate units.

All measurements were normalized to the smallest measurement in the entire dataset including both sexes. Squares of corresponding side lengths (in cm) were drawn in Adobe ILLUSTRATOR. Volume normalization makes the lengths of the sides of squares relative to each other, and so $b$ can be disregarded.

The borders of each square were used to indicate spread in that measurement. The standard errors of the mean were calculated from standard deviations. Their square roots, normalized to the smallest measurement above, were used to determine border thickness (in cm).

royalsocietypublishing.org/journal/rsob    Open Biol. **9**: 190087

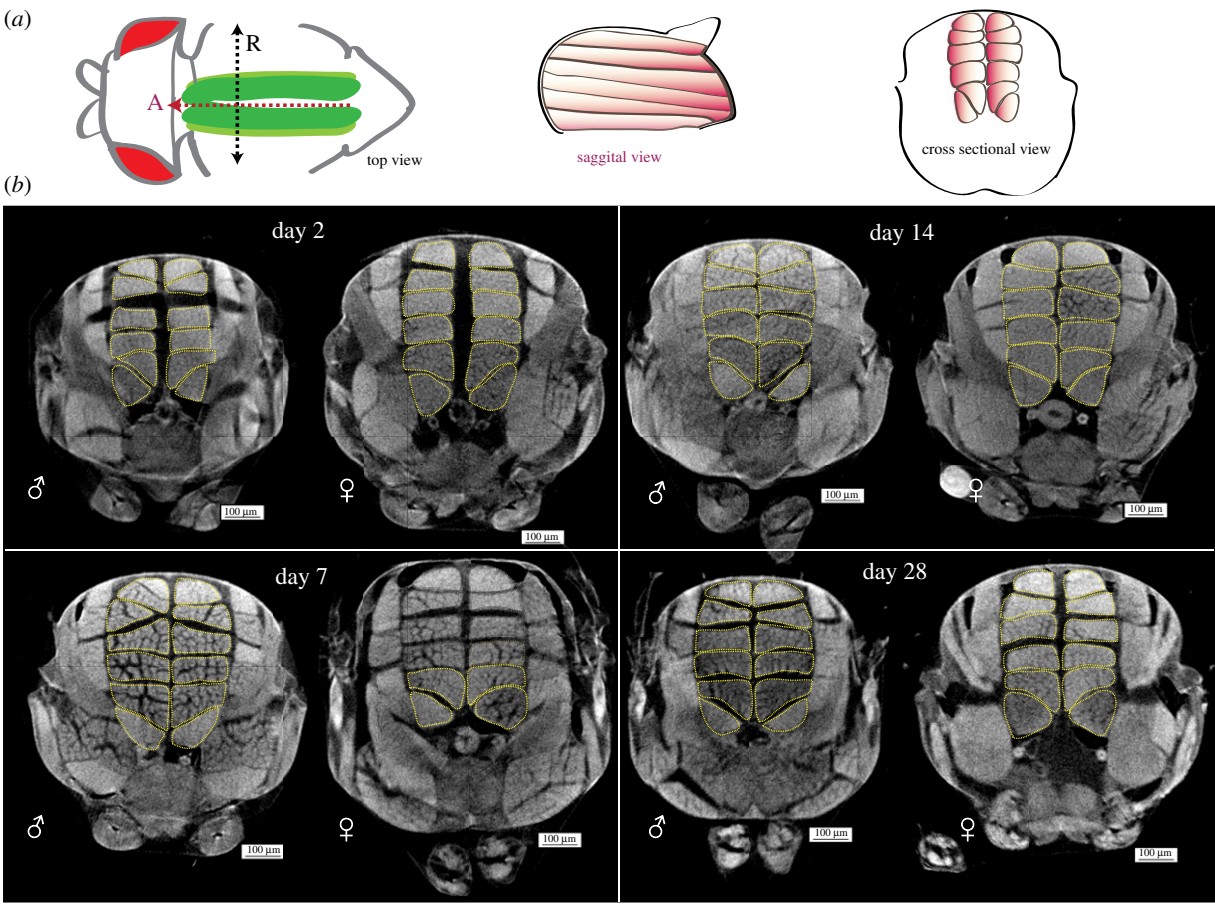

**Figure 1.** Gross changes in male and female adult *Drosophila* longitudinal muscles over time. (*a*) Schematic of dorsal longitudinal muscle (DLM) positioning in the adult thorax. In top view, DLMs (green) run along the A–P axis (red dotted line, arrowhead indicates anterior 'A') inside the thorax, under the cuticle. The black dotted line describes the left–right axis, running between the wing hinges. 'R' denotes the animal's right-hand side. The sagittal view shows six muscle fibres (orange) run anterior to posterior in one hemithorax. In cross-sectional view, six DLMs are arranged in the thorax, on either side of the midline. (*b*) Representative cross sections of whole thorax microCT scans of male (♂) and female (♀) flies at days 2, 7, 14 and 28 post-eclosion. DLMs outlined with yellow dotted lines. Scale bars, 100 μm; *n* = 14–21 per sex per time point.

# 3. Results

## 3.1. microCT scanning of *Drosophila* thoraces reveals muscle structure *in situ*

DLMs are arranged in sets of six muscle fibres, one beneath the other, running anterior to posterior on either side of the sagittal midline, in the dorsal thorax of adult *Drosophila*. Figure 1*a* describes the position of our muscle group of interest, the DLMs, within an adult thorax. Figure 1*b* shows representative thorax cross sections from microCT scans of males and females of 2 to 28 days post-eclosion. Sexual dimorphism in size is obvious from these cross sections.

We employed microCT scanning to visualize thoracic muscles *in situ*. Electronic supplementary material, movie S1 shows a three-dimensional microCT scan thorax of a 2-day-old adult. Surface structures and thoracic musculature can be visualized from these microCT scans. Iodine uptake by myofibrils has previously been demonstrated [14].

The separation of fascicular structures within muscle fibres was intriguing. In electronic supplementary material, movie S2, the separation between myofibrillar bundles running anterior to posterior can be seen in the sagittal view followed by the coronal view. These fascicles diverge along the antero-posterior axis of the muscle and meet at the anterior and posterior ends. This suggests that regions with

signal below the threshold are non-sarcomeric muscle components between fascicles. This arrangement is consistent. We speculate that these non-sarcomeric regions may consist of proteins that define the fascicles, such as those usually present in the extracellular matrix and/or lipids in addition to nuclei, mitochondria and intracellular membranes. Since we have not categorically demonstrated what this volume consists of, we continue to refer to it as non-sarcomeric regions or volume.

Six DLM syncytia on each side of the midline arise during pupariation. Adult muscle progenitors [15] fuse to three remnant syncytia of larval body wall muscles on either side of the midline, called templates. By 20 h apf (after puparium formation), each template splits into two syncytia, giving rise to six fibres in each adult hemithorax [16]. In figure 2*c–e*, we show three animals where templates failed to split during pupariation. These were found among a total of 131 animals scanned for this study. Therefore, the background rate of this specific defect in muscle development in our wild-type stock is roughly 1 in 40 animals.

## 3.2. Mapping individual muscle fibre volumes over time

DLM morphometry was performed using Bruker's CTAN software. In figure 2*a*, DLM nomenclature is described in transverse section. Volumes were measured at 8 μm$^3$ resolution for

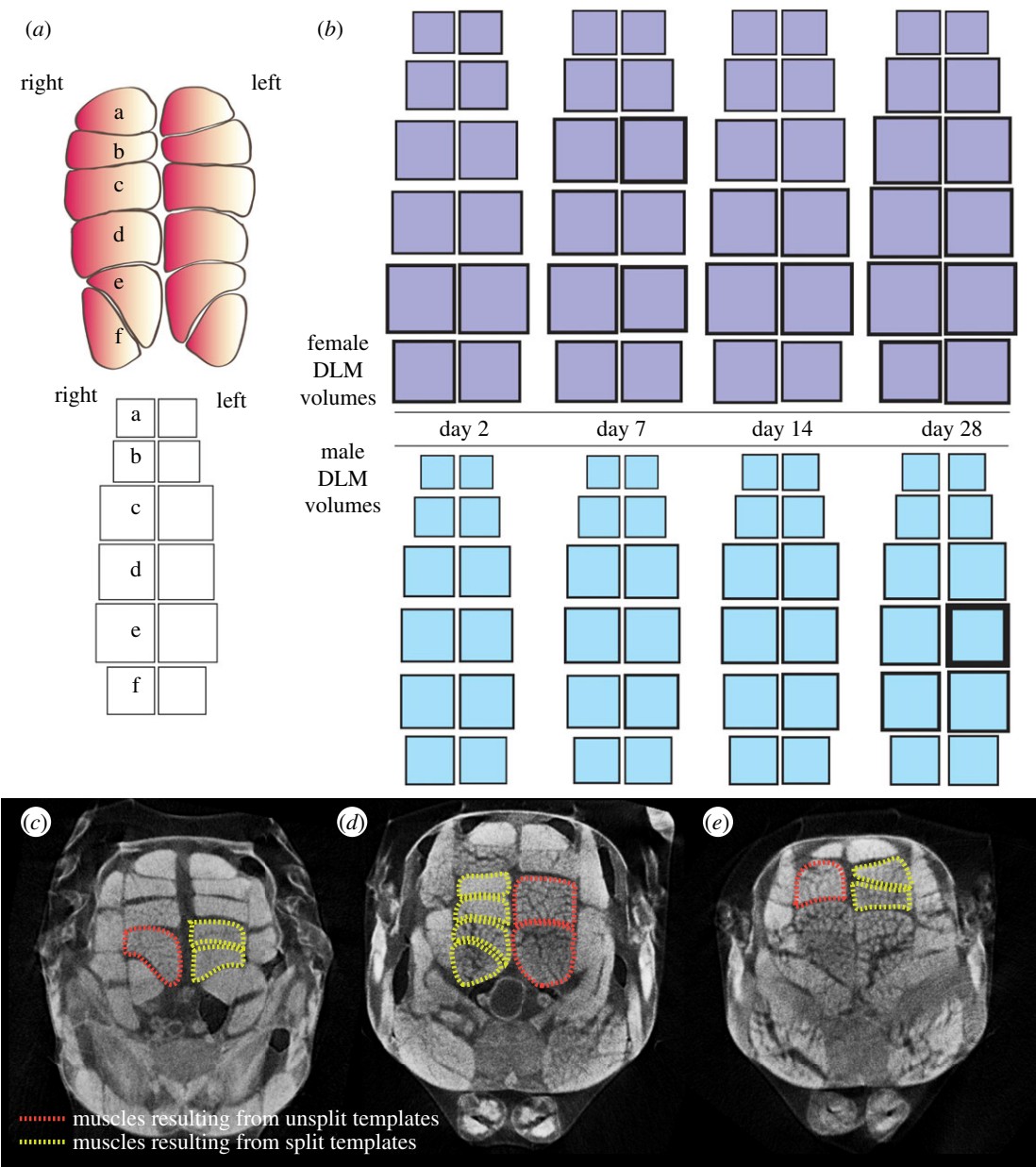

**Figure 2.** DLM volume maps for males and females post-eclosion. (*a*) Schematic of DLM nomenclature. DLM fibres located in the left and right hemithorax are indicated. Individual fibres are marked 'a' to 'f' arranged in a dorsal to ventral manner. The lower panel schematizes a volume map of one full set of DLMs from one animal. The area of each square is proportional to the respective muscle's volume. Each square in the map corresponds to the position of the represented muscle in the thorax. (*b*) Average individual muscle fibre volumes mapped as described in (*a*), for females (purple) and males (blue) at days 2, 7, 14 and 28 post-eclosion. Border thickness for each square represents half the s.e.m. in volume for that muscle, group and time point. $n = 14-21$ animals per sex per time point. (*c*–*e*) Examples of DLM fibres resulting from template splitting defects: cross sections of three different whole thorax microCT scans. Yellow dotted lines outline stereotypically split DLMs, as opposed to those outlined in red dotted lines where template splitting failed during pupariation.

each DLM for males and females at different ages. Average individual DLM volumes and their standard deviations are listed in table 1.

For an uncluttered visual representation of individual muscle placement and normalized volumes within a thorax, we devised a volume mapping protocol. The volume of each muscle fibre is represented by a square of proportional area, arranged according to its positioning in the thorax (figure 2*a*). This representation allows immediate relative comparison of muscle volumes.

For instance, in figure 2*b*, differences in the 'f' muscles in females of any age, and 'd' and 'e' muscles in 28-day-old males, can clearly be seen. This volume mapping protocol may be applied to muscle volumes upon genetic manipulation, physical trauma or for interspecies comparison.

### 3.3. Specific dorsal longitudinal muscles increase in volume differently over time in males and females

Figure 3*a* shows total DLM volumes of females and males at different days post-eclosion. The average total DLM volumes of females are approximately 50% larger than the male counterpart. There is an upward trend in total DLM volumes in both groups over time. Total DLM volumes at day 28 are significantly larger than total DLM volumes at day 2 post-eclosion ($p < 0.001$) in both sexes.

We investigated how the volumes of muscles 'a' to 'f' change over time in males and females. Each animal contributes two volumes of the same muscle from either side of the midline (i.e. $a^L$ and $a^R$ are two independent observations from the same animal at the same time point, with L and R

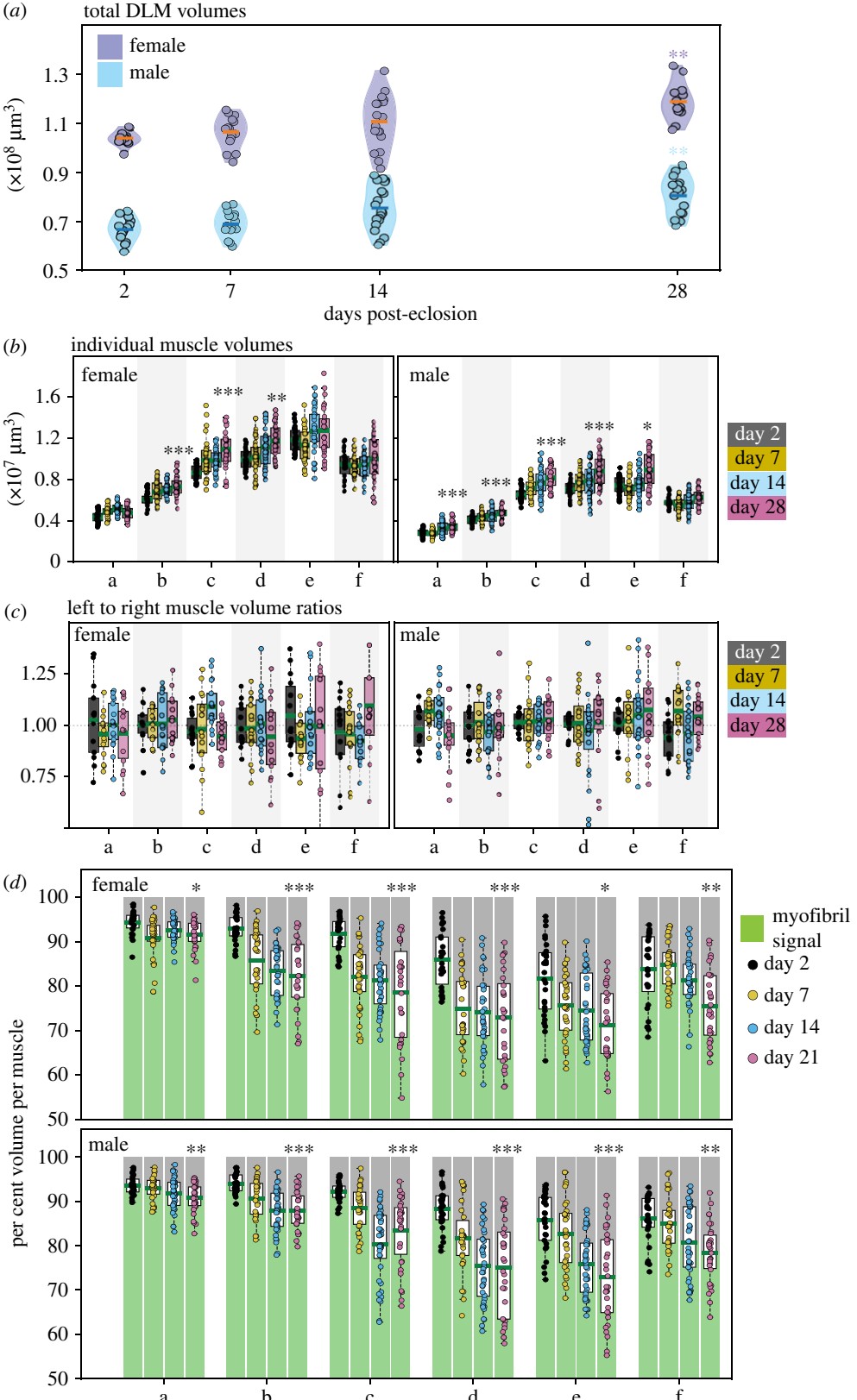

**Figure 3.** Muscle measurements and variation with age. (*a*) Violin plots for total DLM volumes of females (purple) and males (blue) at days post-eclosion (p.e.). Each point denotes an individual observation. Orange and blue bars denote mean total DLM volumes in females and males respectively. (*b*) Box plots for individual muscle volumes for 'a' to 'f' in females (left) and males (right). (*c*) Box plots for left muscle volume to right muscle volume ratios, for muscles 'a' to 'f' in females (left) and males (right), at days 2 (grey), 7 (yellow),14 (blue) and 28 (magenta) p.e. (*d*) Box plots of per cent total volume of each DLM fibre, measured as in (*b*), that is occupied by non-zero signal (myofibril signal, green) and zero signal (grey). Females (top) and males (bottom), for muscles 'a' to 'f', at days 2 (black), 7 (yellow), 14 (blue) and 28 (magenta) p.e. Green bars denote mean myofibril occupied volume per cent, per set. Each point denotes an individual observation. In all plots, $n = 14 - 21$ animals/sex/time point; *, **, *** denote $p < 0.01$, $p < 0.001$, $p < 0.0001$, respectively, while comparing data from day 28 with day 2 using the Wilcoxon rank sum test.

royalsocietypublishing.org/journal/rsob  Open Biol. 9. 190087

**Table 1.** Average individual and total wild-type DLM volumes with corresponding standard deviations at all measured ages for females and males over time.

| | average volume ($\mu m^3$) | | | | s.d. ($\mu m^3$) | | | |
|---|---|---|---|---|---|---|---|---|
| | day 2 | day 7 | day 14 | day 28 | day 2 | day 7 | day 14 | day 28 |
| **females** | | | | | | | | |
| $a^L$ | 4 660 243 | 4 865 094 | 5 159 778 | 4 668 171 | 935 214 | 729 389 | 676 057 | 544 729 |
| $a^R$ | 4 428 057 | 5 002 994 | 5 054 944 | 4 934 014 | 581 075 | 617 658 | 551 736 | 655 401 |
| $b^L$ | 5 912 700 | 6 781 582 | 7 055 889 | 7 438 000 | 1 002 600 | 997 585 | 737 797 | 1 167 168 |
| $b^R$ | 6 042 550 | 7 031 306 | 6 926 728 | 7 497 400 | 717 299 | 1 132 262 | 782 864 | 1 168 091 |
| $c^L$ | 8 610 100 | 10 505 729 | 10 145 372 | 10 903 264 | 766 683 | 3 790 823 | 1 091 784 | 1 727 205 |
| $c^R$ | 8 921 164 | 10 124 612 | 9 502 522 | 11 247 471 | 830 204 | 1 887 063 | 1 226 718 | 2 033 314 |
| $d^L$ | 10 027 986 | 10 187 929 | 11 421 200 | 11 444 068 | 1 240 881 | 1 521 182 | 2 060 164 | 2 375 260 |
| $d^R$ | 10 033 936 | 10 467 247 | 11 110 611 | 12 413 707 | 966 103 | 2 153 768 | 2 020 979 | 2 004 953 |
| $e^L$ | 12 211 643 | 10 701 759 | 12 251 256 | 12 767 843 | 1 628 844 | 2 935 292 | 2 653 295 | 2 653 295 |
| $e^R$ | 11 769 357 | 12 011 571 | 12 746 944 | 13 403 629 | 1 043 190 | 2 279 770 | 2 227 002 | 2 227 002 |
| $f^L$ | 9 417 900 | 9 200 400 | 9 063 628 | 10 468 700 | 1 298 538 | 1 508 968 | 1 288 052 | 1 753 665 |
| $f^R$ | 9 661 257 | 9 433 459 | 10 087 033 | 9 177 000 | 1 798 957 | 1 798 957 | 1 462 736 | 2 588 733 |
| avg. total DLM volume | 101 696 893 | 106 313 682 | 110 525 906 | 115 854 159 | 5 709 912 | 9 653 835 | 11 478 727 | 28 523 041 |
| **males** | | | | | | | | |
| $a^L$ | 2 803 093 | 2 827 638 | 3 387 014 | 3 446 919 | 383 213 | 349 094 | 765 913 | 660 615 |
| $a^R$ | 2 866 127 | 2 752 306 | 3 207 781 | 3 541 513 | 364 171 | 400 067 | 621 885 | 514 545 |
| $b^L$ | 4 094 907 | 4 305 400 | 4 431 438 | 4 842 850 | 488 253 | 529 424 | 809 520 | 751 110 |
| $b^R$ | 4 028 927 | 4 354 088 | 4 613 238 | 4 843 056 | 488 253 | 524 686 | 827 512 | 769 068 |
| $c^L$ | 6 527 500 | 7 346 200 | 8 047 929 | 8 395 881 | 701 170 | 1 023 205 | 1 629 741 | 1 105 114 |
| $c^R$ | 6 504 167 | 7 117 369 | 7 759 990 | 7 846 769 | 770 868 | 906 305 | 1 369 926 | 1 180 071 |
| $d^L$ | 7 167 487 | 7 749 181 | 7 789 614 | 8 714 950 | 936 797 | 1 032 275 | 1 793 874 | 8 714 950 |
| $d^R$ | 7 203 407 | 7 743 688 | 7 994 771 | 8 793 038 | 831 662 | 1 175 831 | 1 706 510 | 1 696 541 |
| $e^L$ | 7 513 287 | 7 384 900 | 7 977 448 | 9 275 094 | 950 729 | 1 394 574 | 1 702 713 | 2 033 750 |
| $e^R$ | 7 341 973 | 7 075 225 | 7 504 938 | 8 466 231 | 782 353 | 803 162 | 1 180 950 | 2 351 929 |
| $f^L$ | 5 634 727 | 5 658 844 | 5 669 319 | 6 321 413 | 525 705 | 568 036 | 984 490 | 759 107 |
| $f^R$ | 6 058 900 | 5 359 344 | 5 980 743 | 6 114 881 | 748 863 | 835 735 | 1 033 071 | 924 268 |
| avg. total DLM volume | 67 744 500 | 74 319 127 | 74 364 224 | 80 602 594 | 5 728 508 | 16 791 698 | 9 613 560 | 7 391 786 |

| number of measurements | females | males |
|---|---|---|
| day 2 | 14 | 15 |
| day 7 | 17 | 16 |
| day 14 | 18 | 21 |
| day 28 | 14 | 16 |

indicating left and right, respectively). Absolute volumes observed from individual muscles are plotted from the two groups in figure 3b.

In figure 3b volumes of muscles 'a' to 'f' at day 2 post-eclosion have been compared with volumes of muscles 'a' to 'f' at day 28 post-eclosion, respectively. Volumes of the muscles 'b' to 'd' from females show statistically significant increases over time, whereas in males, all muscles but 'f' follow this trend. This indicates that individual muscles contribute differently to the increase in total DLM volume with age.

### 3.4. Individual DLM volumes are asymmetric on either side of the midline

To quantify variation between volumes of the same muscle on either side of the midline, we calculated the ratio of left muscle volume to right muscle volume for every muscle fibre ('a' to 'f') in our entire dataset. A left-to-right volume ratio of 1 implies perfect volume symmetry.

Figure 3c shows box plots of individual left-to-right volume ratios for fibres 'a' to 'f', for females and males, at

royalsocietypublishing.org/journal/rsob    Open Biol. 9: 190087

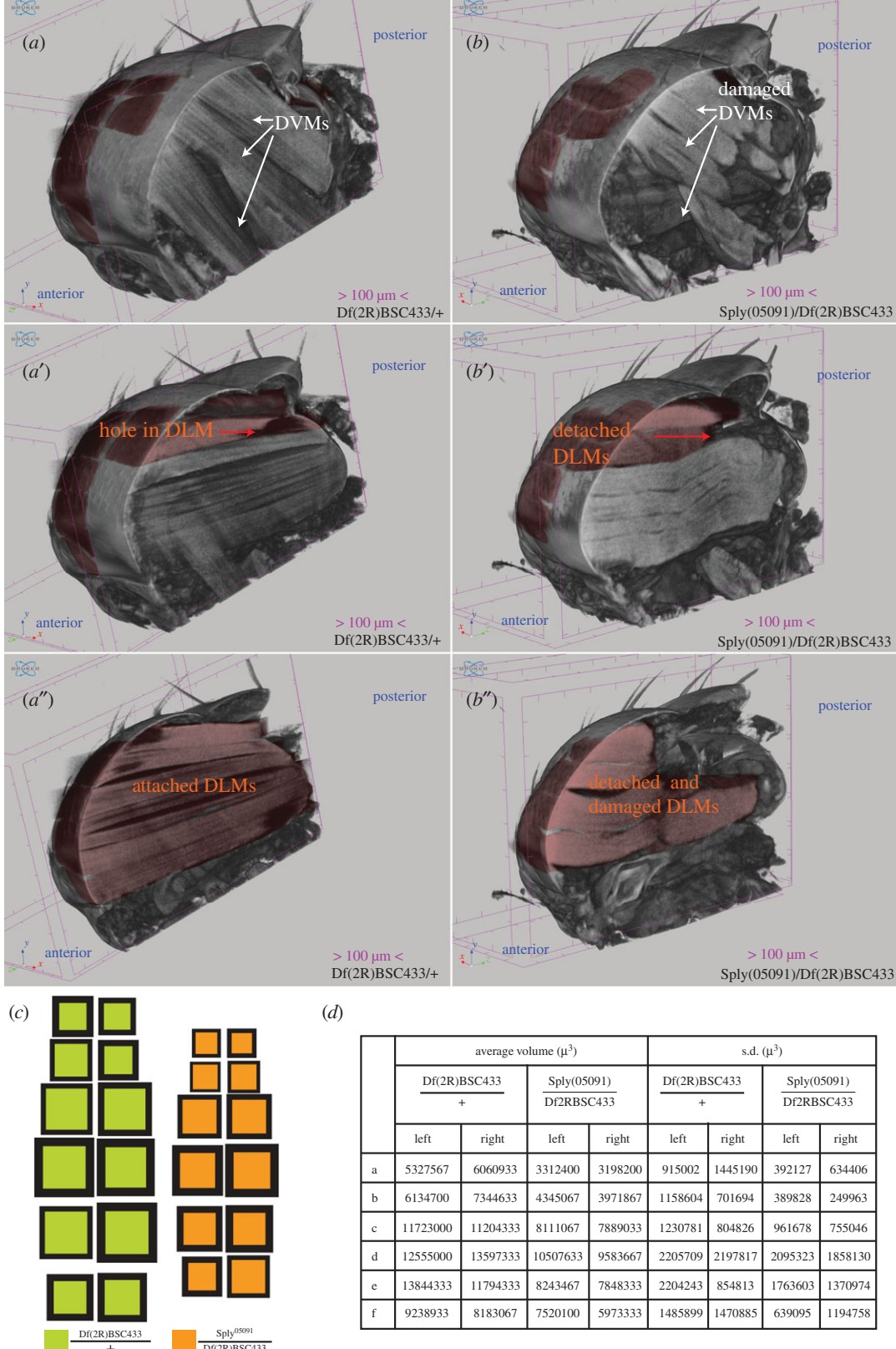

**Figure 4.** DLM phenotypes in *sply* alleles. (*a′,a″*) and (*b′,b″*) show different corresponding sagittal sections of 5-day-old female, single Df(2R)BSC433/+ and sply[05091]/Df(2R)BSC433 thoraces, respectively. In (*a*), white arrows point out control DVM morphology, which is severely compromised in the mutant (*b*). A hole in the b[L] muscle is indicated by the red arrow in (*a′*) whereas in (*b″*) a[L] and b[L] muscle detachment from the cuticle is indicated by the red arrow, though all DLMs in this hemithorax show detachment. DLMs in the right hemithorax are all attached to the cuticle in (*a″*) whereas corresponding DLMs are detached and damaged in (*b″*). (*c*) Volume maps for each genotype. (*d*) Average DLM volumes and s.d. in each group described above. $n = 3$ per group.

different times in adulthood. The average left to right muscle volume ratios, for all DLMs, in both groups tend towards unity. However, the spread of individual observations in

these sets ranges from below 0.75 to above 1.25. Therefore, this fluctuation in contralateral muscle volumes should be recognized while using them as internal controls.

## 3.5. Non-sarcomeric volumes in muscle fibres increase over time

We examined whether there was a consistent trend in the volume separating the fascicles over time. Thus, we plotted the percentage of volume occupied by signal voxels (myofibril signal) in the three-dimensional region bounding each muscle.

In figure 3*d*, we see a consistent downward trend in these per cent volume fractions occupied by myofibril signal in a muscle fibre. The volume fractions occupied by myofibril signal in muscles is reduced significantly with time in all muscles in both sexes, ranging from 5% to 15%. This consistent trend suggests this observation is unlikely to be an artefact and is likely to be a bonafide signature of ageing.

The molecular, cellular and functional significance of this age-dependent alteration remains to be investigated.

## 3.6. Assessing defects in adult DLMs in *Drosophila* mutants

We have documented morphological variation in wild-type DLMs post-eclosion into 28 days of adulthood so far. In our wild-type sample set, DLMs are largely uniform structures. However, in genetic conditions where DLMs are known to be fragile [17–19], we would greatly benefit from recording the exact nature of muscle defects beyond single planes, at scale.

To demonstrate this application, we performed DLM morphological analysis of mutants of *sply*. Sply (Sphingosine-1-phosphate Lyase, CG8946) has documented roles in muscle maintenance [18,20,21]. We tested a loss of function allele of *sply*$^{05091}$ [20] over the deficiency Df(2R)BSC433 in trans and the Df(2R)BSC433 heterozygote as a control. In figure 4*a,a',a''* we show three planes through the same control thorax. DLMs coloured in red are to be compared with corresponding muscles in the mutant thorax. All DLMs appear attached to the exoskeleton, with the exception of 'a' muscles, which are damaged at the anterior end. Also, b$^L$ has a hole at the posterior end, suggesting this deficiency is a dominant allele. In figure 4*b,b',b''*, we show three corresponding planes through the same mutant thorax. Clearly, all DLMs in this thorax are detached at the posterior end and considerable additional damage is visible in DLMs in the right hemithorax. The entire comparison can also be seen in electronic supplementary material, movie S3 in three dimensions.

The volume maps in figure 4*c* show the variation in muscle volumes, clearly showing that *sply* alleles in trans result in reduced muscle volume compared to the deficiency heterozygote. Muscle volumes and their standard deviations are listed in figure 4*d*.

## 3.7. Method applies to other insect species

We tested our protocols on honeybee thoraces. Electronic supplementary material, movie S4 shows a microCT scan of an *Apis dorsata* thorax from the NCBS apiary. The DLM group has been segmented out and clearly differs vastly in fibre number and shape from *Drosophila* DLMs.

The protocol was changed for this prep: I$_2$/KI solution incubation for 72 h allowed sufficient diffusion into the honeybee thorax. With appropriate variations in incubation times and I$_2$/KI volumes, this protocol may apply to many other species. We hope that staining protocol and variations thereof will assist accurate *in situ* reporting of internal soft tissue across insect species.

## 4. Discussion and conclusion

In addition to assessing function, variations in tissue morphology through the processes of development, ageing, disease and repair are critical to fully understanding organism function. Muscle function is key to quality of life, survival and metabolic regulation in many species [22–24]. DLMs model homeostatic adult muscles with a fibrillar arrangement that is shared with mammalian skeletal muscles. The vast genetic toolkit, short lifespan and shared mechanisms of adult repair make *Drosophila* DLMs a promising model of adult human muscle repair and pathologies [25,26].

Through a contrasting regime in the near isotonic aqueous-based medium in microCT scanning, we are able to measure *in situ Drosophila* muscle morphology and arrangement, at a significantly improved combination of scale and resolution. We have also elucidated the variations in individual adult fruit fly DLM morphology in an ageing and sex-dependent manner.

Our results have shone a light on aspects of DLM ageing that could not be satisfactorily interrogated otherwise. New avenues have arisen based on these data. For instance, the volume between fascicles that increases with age may be constituted by extracellular matrix proteins. In mammalian contexts, muscle stiffness and extracellular matrix protein deposition increase with age [27,28]. Our volume measurements encourage investigations along these lines and exploring the role of ECM in age-related flight performance decrement. In addition, which factors stimulate differential muscle growth in DLMs is a completely new question. Also, with measurements of this accuracy, the question of mechanisms that coordinate muscle size symmetry can be addressed.

Further, in myopathy models, muscle distortions contributed by the dissection process can be entirely avoided and *in situ* size and shape can be measured accurately. We have demonstrated this through our examination of the published DLM phenotype of *sply* loss of function alleles [18,20]. Also, it is now possible to study the morphology of relatively inaccessible muscles.

The focus of this study has been morphological correlates of DLM ageing in *Drosophila*. This method may find application in studying other *Drosophila* soft tissues like the eye, gut and heart. The accuracy and ease of this protocol coupled with automated analysis promises to find applications in preliminary genetic and pharmaceutical screens looking for systemic effects at scale.

Clearly, this protocol has value in comparative studies across species. A wealth of the world's insect biodiversity still remains to be explored. Accurate morphological and anatomical records will supplement genetic analyses in intra-species and inter-species evolution. For instance, precise measurements of musculature in different castes of ant and honeybee colonies may further inform investigations into the molecular details of their development [29]. In all, this technique promises to be a valuable addition to the toolkit of biology.

Ethics. This study satisfies the requirements of the safety and ethics committee at NCBS-TIFR.

Data accessibility. All data supporting this article have been uploaded as part of the manuscript or electronic supplementary material.

Authors' contributions. D.C. conceptualized and designed the study, acquired, analysed and interpreted data, drafted and revised the manuscript; S.P. processed the raw data and helped standardize

the protocol; A.A. plotted the data in Python; K.B.A. prepared the *sply* samples; K.V. assessed and commented on the data, provided intellectual and biological context related input and approved the final version of the manuscript for publication.
Competing interests. We have no competing interests.

Funding. Funding for this study was provided by NCBS and the JC Bose fellowship to K.V. D.C. is funded by NCBS's Campus Fellows Program. S.P. was funded by NCBS. A.A. was funded by CSIR, India.
Acknowledgements. We acknowledge Axel Brockman Lab for bees from their colony at NCBS. Devam Purohit helped with annotation of the videos.

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
