## [Reviewer comments · Open Biology]

Review History

RSOB-19-0087.R0 (Original submission)

Review form: Reviewer 1

Recommendation

Major revision is needed (please make suggestions in comments)

Are each of the following suitable for general readers?

- a) **Title**
Yes

- b) **Summary**
No

- c) **Introduction**
No

Is the length of the paper justified?

Yes

Should the paper be seen by a specialist statistical reviewer?

No

Is it clear how to make all supporting data available?

Yes

Is the supplementary material necessary; and if so is it adequate and clear?

Yes

Do you have any ethical concerns with this paper?

No

Comments to the Author

This manuscript successfully applies the technique of MicroCT scanning to study the structure of direct longitudinal muscles in *Drosophila*, as well as in honeybee. Some new insights into muscle structure and development are presented here, but overall this is a methodology paper. As the authors point out, MicroCT will provide better appreciation of structure of mutant muscles in *Drosophila*, as well as thoracic muscles in other insects. The main point for consideration is whether the method will be of enough interest to warrant publication in *Open Biology*. If the authors wish to convert the paper into a research report, specific hypotheses about structural defects in mutant muscles could be tested and reported upon. At the least, questions addressed in the manuscript as to differences in muscle volumes between the sexes and during aging could be highlighted in the abstract and introduction, so that there is some focus on research observations as opposed to only methodology. Below are some specific comments that could improve the presentation:

- 1) References need to be added in multiple locations, e.g., “shares myofibrillar architecture with mammalian skeletal muscles”; “*Drosophila* muscle developmental studies focus on larvae or pupae”
- 2) The authors focus on DLMS and note that they alone share architecture with mammalian muscle. What about DVMs? Don't the authors mean to talk about IFMs overall? Also, some explanation of how these muscle work to power flight would be appropriate.
- 3) The single paragraph sentence at the end of the Introduction needs to be better developed to highlight the research results. The sentence prior to that should be fused with the previous paragraph. The same for line 104.
- 4) How long does a scan take? What is the purpose of calculating square roots of muscle volumes?
- 5) Lines 108-109: “actin bundles” are more properly called “myofibrillar bundles” or perhaps “fascicles”?
- 6) Line 117: Fig. 2C-E seems to show four instances of templates failing to split, not three. Need to explain how the rate of this defect was calculated (1:40).
- 7) Line 126: we devised a volume mapping [protocol]
- 8) Unclear what is meant by non-myofibril signal. Just the area between fascicles? What about

signals due to mitochondria, nuclei and intracellular membranes? Actin signal in Fig 3D is either referring to myofibrillar signal or fascicle signal, depending on what is meant here. This should be clarified.

9) A number of grammatical errors are sprinkled through the manuscript. Note author capitalization or repetition errors in references 8 and 10.

10) Figure 2B caption mentions 28-day results, whereas the figure says 3 weeks. In addition to correcting this, it is probably best to use either days or weeks in both figure and legend.

Review form: Reviewer 2

Recommendation

Accept as is

Are each of the following suitable for general readers?

- a) **Title**
Yes
- b) **Summary**
Yes
- c) **Introduction**
Yes

Is the length of the paper justified?

Yes

Should the paper be seen by a specialist statistical reviewer?

No

Is it clear how to make all supporting data available?

Yes

Is the supplementary material necessary; and if so is it adequate and clear?

Yes

Do you have any ethical concerns with this paper?

No

Comments to the Author

This is an excellent alternative to sectioning and the text is clear and concise. I highly recommend publication.

Decision letter (RSOB-19-0087.R0)

07-May-2019

Dear Dr Chaturvedi,

We are pleased to inform you that your manuscript RSOB-19-0087 entitled "Adult *Drosophila* Muscle Morphometry through MicroCT reveals dynamics during aging" has been accepted by the Editor for publication in *Open Biology*. The reviewer(s) have recommended publication, but also suggest some minor revisions to your manuscript. Therefore, we invite you to respond to the reviewer(s)' comments and revise your manuscript.

Please submit the revised version of your manuscript within 7 days. If you do not think you will be able to meet this date please let us know immediately and we can extend this deadline for you.

- 1) A text file of the manuscript (doc, txt, rtf or tex), including the references, tables (including captions) and figure captions. Please remove any tracked changes from the text before submission. PDF files are not an accepted format for the "Main Document".
- 2) A separate electronic file of each figure (tiff, EPS or print-quality PDF preferred). The format should be produced directly from original creation package, or original software format. Please note that PowerPoint files are not accepted.
- 3) Electronic supplementary material: this should be contained in a separate file from the main text and meet our ESM criteria (see <https://royalsocietypublishing.org/rsob/for-authors>). All supplementary materials accompanying an accepted article will be treated as in their final form. They will be published alongside the paper on the journal website and posted on the online figshare repository. Files on figshare will be made available approximately one week before the accompanying article so that the supplementary material can be attributed a unique DOI.

Online supplementary material will also carry the title and description provided during submission, so please ensure these are accurate and informative. Note that the Royal Society will not edit or typeset supplementary material and it will be hosted as provided. Please ensure that the supplementary material includes the paper details (authors, title, journal name, article DOI). Your article DOI will be 10.1098/rsob.2016[last 4 digits of e.g. 10.1098/rsob.20160049].

4) A media summary: a short non-technical summary (up to 100 words) of the key findings/importance of your manuscript. Please try to write in simple English, avoid jargon, explain the importance of the topic, outline the main implications and describe why this topic is newsworthy.

Images

Data-Sharing

It is a condition of publication that data supporting your paper are made available. Data should be made available either in the electronic supplementary material or through an appropriate repository. Details of how to access data should be included in your paper. Please see <https://royalsocietypublishing.org/rsob/for-authors#question5> for more details.

Data accessibility section

Sincerely,

The Open Biology Team

<mailto:openbiology@royalsociety.org>

Board Member

Comments to Author:

The authors make a convincing argument that new methods for analyzing muscles are needed, and their use of MicroCT scans reveals a powerful approach to evaluate tissue in the context of the whole organism. Their data clearly show differences in muscle volume between sexes, in part through elegant 3D movies, and their method is applicable to another insect species, honeybee. The impact of this work would be significantly higher, however, if the analyses included a description of a muscle mutant.

In addition to the other issues noted by one of the reviewers, minor weaknesses that need attention include:

- 1) grammar and punctuation;
- 2) justification for measurements to two decimal places (not warranted by the standard deviations);
- 3) in Fig 1B, is anterior down; are we seeing antennae?
- 4) minor annotation on the movies to show features.

Reviewer(s)' Comments to Author:

Referee: 1

Comments to the Author(s)

This manuscript successfully applies the technique of MicroCT scanning to study the structure of direct longitudinal muscles in *Drosophila*, as well as in honeybee. Some new insights into muscle structure and development are presented here, but overall this is a methodology paper. As the authors point out, MicroCT will provide better appreciation of structure of mutant muscles in *Drosophila*, as well as thoracic muscles in other insects. The main point for consideration is whether the method will be of enough interest to warrant publication in *Open Biology*. If the authors wish to convert the paper into a research report, specific hypotheses about structural defects in mutant muscles could be tested and reported upon. At the least, questions addressed in the manuscript as to differences in muscle volumes between the sexes and during aging could be highlighted in the abstract and introduction, so that there is some focus on research observations as opposed to only methodology. Below are some specific comments that could improve the presentation:

- 1) References need to be added in multiple locations, e.g., “shares myofibrillar architecture with mammalian skeletal muscles”; “*Drosophila* muscle developmental studies focus on larvae or pupae”
- 2) The authors focus on DLMs and note that they alone share architecture with mammalian muscle. What about DVMs? Don't the authors mean to talk about IFMs overall? Also, some explanation of how these muscle work to power flight would be appropriate.
- 3) The single paragraph sentence at the end of the Introduction needs to be better developed to highlight the research results. The sentence prior to that should be fused with the previous paragraph. The same for line 104.
- 4) How long does a scan take? What is the purpose of calculating square roots of muscle volumes?
- 5) Lines 108-109: “actin bundles” are more properly called “myofibrillar bundles” or perhaps “fascicles”?
- 6) Line 117: Fig. 2C-E seems to show four instances of templates failing to split, not three. Need to explain how the rate of this defect was calculated (1:40).
- 7) Line 126: we devised a volume mapping [protocol]
- 8) Unclear what is meant by non-myofibril signal. Just the area between fascicles? What about signals due to mitochondria, nuclei and intracellular membranes? Actin signal in Fig 3D is either referring to myofibrillar signal or fascicle signal, depending on what is meant here. This should be clarified.
- 9) A number of grammatical errors are sprinkled through the manuscript. Note author capitalization or repetition errors in references 8 and 10.
- 10) Figure 2B caption mentions 28-day results, whereas the figure says 3 weeks. In addition to correcting this, it is probably best to use either days or weeks in both figure and legend.

Referee: 2

Comments to the Author(s)

This is an excellent alternative to sectioning and the text is clear and concise. I highly recommend publication.

Author's Response to Decision Letter for (RSOB-19-0087.R0)

See Appendix A.

Decision letter (RSOB-19-0087.R1)

30-May-2019

Dear Dr Chaturvedi

We are pleased to inform you that your manuscript entitled "Adult *Drosophila* Muscle Morphometry through MicroCT reveals dynamics during aging" has been accepted by the Editor for publication in Open Biology.

Sincerely,

The Open Biology Team
mailto:openbiology@royalsociety.org

Appendix A

Editor,
Royal Society Open Biology.

26th May 2019

Subject: Submission of revised Manuscript

Dear Editor,

The authors of this study are very happy to have received your referees' comments on our manuscript. They were constructive and we believe they have led to an improved paper. We have addressed each comment specifically. In the following pages, you'll find that the text of each specific comment from the reviewers has been changed to red and our response that describes consequent changes to the manuscript is in blue. We sincerely hope that these revisions are to your satisfaction.

This manuscript was intended to illustrate the power of using MicroCT scanning for analysing muscles in insects. These novel anatomical observations correlated with *Drosophila* ageing couldn't have satisfactorily been made otherwise. In rewriting the manuscript we have tried to make it hypothesis driven, as Reviewer 1 suggested. We are convinced that this method and these findings stand on their own, irrespective of our success in converting the original submission to a hypothesis driven paper. We hope that you agree.

We deeply appreciate the quick turn around on our submission. If there is any other information that we need to furnish, we would be happy to provide it.

Sincerely,
Dhananjay Chaturvedi

Board Member

Comments to Author:

The authors make a convincing argument that new methods for analyzing muscles are needed, and their use of MicroCT scans reveals a powerful approach to evaluate tissue in the context of the whole organism. Their data clearly show differences in muscle volume between sexes, in part through elegant 3D movies, and their method is applicable to another insect species, honeybee. **The impact of this work would be significantly higher, however, if the analyses included a description of a muscle mutant.**

We are grateful to the Board member for recognizing the need and scope of MicoCT scanning of *Drosophila* muscles. Through figure 4, Video 4 and its associated result description, we have added a small analysis of the Indirect Flight Muscle phenotypes mutations in the Sphingosine Lyase gene. This mutation and its effects on *Drosophila* DLMs have been previously described (Pantoja et. al. Development. 2013). This permits for a direct comparison between traditional methods and ours. A full aging analysis on these mutants would prove impractical, given the viability of these mutants and the scope of this study.

In addition to the other issues noted by one of the reviewers, minor weaknesses that need attention include:

1) grammar and punctuation;

The revised manuscript has been checked for spelling and grammar.

2) justification for measurements to two decimal places (not warranted by the standard deviations);

We thank the review for this comment. The least count in our measurements is above $1\mu^3$ and they range from the fifth to the 7th order of magnitude. Therefore, we have rounded off volume and standard deviation measurements to the nearest whole number.

3) in Fig 1B, is anterior down; are we seeing antennae?

Fig1B. schematizes the positioning of DLMs inside the thorax. In the legend for this figure, we have described the Top view as: "In Top view, DLMs (Green) run along the A-P (red dotted line, arrowhead indicates anterior 'A') inside the thorax, under the cuticle." To illustrate this positioning to the reader, the head with eyes and antennae are shown. We hope this, in addition to the labelling, communicates to the reader that this is in fact, the top view of the animal with the anterior side to the left, and therefore the posterior to the right of the page.

4) minor annotation on the movies to show features.

All uploaded videos are now annotated to guide the viewer. They highlight structures that we'd like to draw the viewer's attention to.

Reviewer(s)' Comments to Author:

Referee: 1

Comments to the Author(s)

This manuscript successfully applies the technique of MicroCT scanning to study the structure of direct longitudinal muscles in *Drosophila*, as well as in honeybee. Some new

insights into muscle structure and development are presented here, but overall this is a methodology paper. As the authors point out, MicroCT will provide better appreciation of structure of mutant muscles in *Drosophila*, as well as thoracic muscles in other insects. The main point for consideration is whether the method will be of enough interest to warrant publication in Open Biology. If the authors wish to convert the paper into a research report, specific hypotheses about structural defects in mutant muscles could be tested and reported upon. **At the least, questions addressed in the manuscript as to differences in muscle volumes between the sexes and during aging could be highlighted in the abstract and introduction, so that there is some focus on research observations as opposed to only methodology.**

We thank the reviewer for suggesting these changes. The abstract, introduction and discussion have been extensively modified bearing this comment in mind.

Below are some specific comments that could improve the presentation:

1) References need to be added in multiple locations, e.g., “shares myofibrillar architecture with mammalian skeletal muscles”; “*Drosophila* muscle developmental studies focus on larvae or pupae”

We thank the reviewer for this suggestion. We have added more references as appropriate. The manuscript now provides better context to our findings.

2) The authors focus on DLMs and note that they alone share architecture with mammalian muscle. What about DVMs? Don't the authors mean to talk about IFMs overall? Also, some explanation of how these muscle work to power flight would be appropriate.

This description has been added to the introduction.

3) The single paragraph sentence at the end of the Introduction needs to be better developed to highlight the research results. The sentence prior to that should be fused with the previous paragraph. The same for line 104.

We thank the reviewer for this suggestion. The two sentences indicated by the reviewer have been fused to previous paragraphs.

4) How long does a scan take? What is the purpose of calculating square roots of muscle volumes?

Each scan takes forty minutes. This information has been added to the **MicroCT scanning** section under materials and methods.

The reason for calculating square roots of the volume lies in the design of the volume mapping protocol. We wanted to represent the magnitude of volume (3D) through the area of a square (2D). So, if a square of side length 's' represents a volume 'V'

$$s^2 = bV$$

where 'b' is a constant of appropriate units, the length 's', of the side of the representative square, is square root of bV. Because these volume measurements are normalized to the smallest measurement, the constant 'b' cancels out. This clarification is very warranted and so has been added to the manuscript in the **Data processing and volume calculation** section under Materials and Methods.

5) Lines 108-109: “actin bundles” are more properly called “myofibrillar bundles” or perhaps “fascicles”?

We thank the reviewer for this suggestion. “actin bundles” have been changed to “myofibrillar bundles”.

6) Line 117: Fig. 2C-E seems to show four instances of templates failing to split, not three. Need to explain how the rate of this defect was calculated (1:40).

We thank the reviewer for pointing out this possible source of confusion. To clarify that statement, we’ve amended the text as below.

“...we show three **animals** where templates failed to split during pupariation. **These were found among a total of 131 animals scanned for this study.** Therefore the background rate of this specific defect in muscle development in our wildtype stock is around 1 in 40 **animals.**”

7) Line 126: we devised a volume mapping [protocol]

This correction has been made in line 126 and 130 of the original submission. The title of Figure 2 has been amended in the same spirit.

8) Unclear what is meant by non-myofibril signal. Just the area between fascicles? What about signals due to mitochondria, nuclei and intracellular membranes? Actin signal in Fig 3D is either referring to myofibrillar signal or fascicle signal, depending on what is meant here. This should be clarified.

Based on comparisons with phalloidin staining of DLM preps, and that iodine is taken up by myofibrils, we are confident, that the signal we are seeing in our scans is associated with myofibrils. To highlight this, we have moved the sentence “Iodine uptake by myofibrils has previously demonstrated (6).” from the Materials and methods to the second paragraph in the first result section.

Since we are not absolutely certain what the remaining muscle volume is occupied by, we have stated the following in the next paragraph “This suggests that regions with signal below threshold, are **non-sarcomeric** muscle components between fascicles. This arrangement is consistent. We speculate these **non-sarcomeric** regions may consist of proteins that define the fascicles, such as those usually present in the extracellular matrix and/or lipids in addition to nuclei, mitochondria and intracellular membranes. Since we have not categorically demonstrated what this volume consists of, we continue to refer to it as **non-sarcomeric** regions or volume.” To avoid confusion, “Actin bundles” labeled next to the green box in Fig.3D has been changed to “Myofibril signal”.

9) A number of grammatical errors are sprinkled through the manuscript. Note author capitalization or repetition errors in references 8 and 10.

We attended to all such errors in this manuscript. We thank the reviewer for their attention to detail.

10) Figure 2B caption mentions 28-day results, whereas the figure says 3 weeks. In addition to correcting this, it is probably best to use either days or weeks in both figure and legend.

We thank the reviewer for pointing out this error in figure-making. It has been duly corrected.

Referee: 2

Comments to the Author(s)

This is an excellent alternative to sectioning and the text is clear and concise. I highly recommend publication.

This comment is deeply appreciated and we thank the reviewer for it.